# Development of a Comprehensive Process for Introducing Game-Based Learning in Higher Education for Lecturers

María Fernández-Raga [1,*], Darija Aleksić [2,*], Aysun Kapucugil İkiz [3], Magdalena Markiewicz [4] and Herbert Streit [5]

1   Department of Applied Physics, University of León, 24071 León, Spain
2   School of Economics and Business, University of Ljubljana, 1000 Ljubljana, Slovenia
3   Faculty of Business, Dokuz Eylul University, 35390 Izmir, Turkey
4   Faculty of Economics, University of Gdansk, Bazynskiego 8, 80-309 Gdansk, Poland
5   Faculty of Business and Transport Management, Heilbronn University of Applied Sciences, 74081 Heilbronn, Germany
*   Correspondence: maria.raga@unileon.es (M.F.R.); darija.aleksic@ef.uni-lj.si (D.A.); Tel.: +34-98-729-1000 (ext. 5342) (M.F.R.)

**Abstract:** Emerging trends such as digitalization, globalization, and the COVID-19 pandemic are forcing higher education institutions to undergo constant organizational and technological changes and to introduce innovative pedagogical approaches suitable for teaching a new generation of students—the so-called digital natives. The goal of this paper is to engage in the ongoing debate in higher education about new teaching methods, i.e., game-based learning methods, which meet the needs of digital natives. They have grown up in a fast-paced, technology-driven society, which has affected how they absorb information, their ability to concentrate for extended periods, and their motivation and engagement in the learning process. Existing research suggests that implementing the game-based learning method can be very difficult and costly, as it often requires adapting the freely available game to the requirements of the particular course and additional investment in purchasing appropriate equipment. In this paper, we develop a comprehensive procedure for introducing a cost-effective game-based learning method in higher education, which includes thirteen steps to help lecturers introduce game-based activities straightforwardly into their teaching processes. In addition, we also present security, cultural, and quality assurance issues that need to be considered when implementing game-based learning in higher education.

**Keywords:** game-based learning; gamification; game design; higher education; hybrid teaching; online teaching

## 1. Introduction

High-quality education is crucial to ensure social cohesion, competitiveness, and sustainable growth in a changing, digitalized society and globally competitive environment. To meet the demands of rapid change in the world and prepare students for the job market of the future, as described by institutions such as the European Commission and the literature [1,2], higher education institutions (HEIs) have embraced the trend of digital transformation [3]. The COVID-19 outbreak in 2020 has further accelerated the existing trend toward the digitalization of higher education [3–5] and increased the need for HEIs to leverage the transformative benefits of information and communication technologies (ICTs) to enrich teaching, enhance the learning experience, facilitate access to higher education through distance learning, and promote internationalization [6–9].

In line with Sustainable Development Goal 4, quality education, which requires, among other things, that youths and adults have relevant skills for decent work, the digitalization of higher education is necessary not only to meet the new demands of the job market [5] but also to meet the needs of the digital natives (i.e., the new generation in

higher education), who have unique characteristics of the digital age and are known as the most diverse generation we have ever had to teach. Digital natives born after 1980 [10,11], are more tech-savvy than any previous generation [12]. As digital technologies such as computer hardware, software, smartphones, the Internet, and networks are an integral part of their everyday lives [13], digital natives expect the same level of technology in their learning environment as they do in their lives [14,15]. As they interact with the user interface elements of these technologies, they recognize the value of the technology after a certain period of use, and their user experience may change from negative to positive over time [4]. Research also shows that they have a decreased ability to pay constant cognitive attention [16]. Therefore, educators should follow modern educational trends and use different teaching methods and approaches enhanced by ICT to motivate and engage digital native students in the learning process [17].

One of the innovative learning approaches that can help educators address the needs, preferences, and demands of digital natives is game-based learning (GBL), defined as "an environment where game content and gameplay enhance knowledge and skills acquisition, and where game activities involve problem-solving spaces and challenges that provide players/students with a sense of achievement" [18]. According to Pho and Dinscore [19], GBL involves developing games for students and designing learning activities that can incrementally introduce concepts and guide users toward an end goal. GBL leans on game principles and applies them to real-life scenarios to encourage students to engage in learning through play and make the learning process more interesting by making it fun [20]. GBL and gamification, defined as the use of game design elements (e.g., points, penalties, leaderboards, and trophies) in a traditionally non-game context to influence behavior, are often referred to as similar concepts. Still, most researchers argue that they are related but distinct concepts [21]. The biggest difference between the concepts is that GBL uses a game as part of the learning process and turns a single learning objective into a game (i.e., using games to achieve learning outcomes), while gamification turns the learning process as a whole into a game [20]. Gamification in education is used to support the learning material, while game-based learning is a way to learn and achieve learning objectives through games.

Some of the existing research suggests that GBL positively impacts many outcomes relevant to the educational context, such as positively impacting attitudes and perceptual and cognitive skills [22], increasing students' motivation to learn and engage, providing opportunities for exploration and acquisition of new knowledge and skills [18,23], and stimulating students to more easily achieve and maintain undivided attention for a longer period [20]. GBL is fun and thus increases students' interest and motivation in the subject. In addition, GBL activities are different from regular curriculum activities and, therefore, may be perceived as a mechanism for disrupting routine, which may increase students' attention and interest in absorbing new information. On the other hand, a review of the literature on game-based learning also revealed that some studies had not found clear evidence of a positive relationship between GBL and students' high academic achievement or psychological development, suggesting that GBL may not be more effective than traditional classroom lectures [18].

Although GBL has been introduced in various fields (e.g., computer science, biology, business, logistics, mathematics, physics, psychology, and statistics), the use of this innovative teaching method still needs to be improved in practice. According to Al-Azawi et al., the introduction of GBL is quite expensive and difficult [20]. High-quality development requires many resources and knowledge about how to design GBL in an interesting way for students. If GBL is supported by ICT, good access is required, and students and lecturers must be equipped with adequate digital skills. It is also quite difficult to integrate course content and learning objectives, especially in technical subjects, into interesting and engaging storytelling on which GBL is based. In addition, learning progress, as measured by academic integration, social integration, performance, and students' persistence in learning, is critical to students' success [17]. Due to the characteristics and speed of implementation of GBL activities, lecturers may have limited insight into students' progress and, therefore

restricted ability to create feedback for students, resulting in students not excelling in the GBL activity or even becoming stuck along the way without the lecturer noticing [17]. Moreover, o According to Pho and Dinscore [19], GBL involves developing games for students and designing learning activities that can incrementally introduce concepts and guide users toward an end goal.ne of the important reasons for the limited diffusion of GBL in the educational setting is the fact that the literature in the field is very fragmented, making it very time-consuming for lecturers to become familiar with the method to integrate it into the teaching process. A recent study also showed that lecturers often do not have the time and resources to do additional work to acquire, learn, and teach educational games [24].

Building on existing literature and models (e.g., the Game-Based Learning Design Model by Shi and Shih [25]), this paper aims to provide comprehensive, step-by-step guidance on how to implement GBL into course didactics in a systematic, cost-and-time-efficient manner. Specifically, in this paper, we describe in detail the processes and steps of a comprehensive implementation of game-based learning suitable for lecturers of social and science courses new to the field and have limited resources available. In addition, we highlight security, protection, and cultural issues that need to be considered when implementing GBL, as well as a process-oriented quality assurance model for the GBL method in higher education. Thereby, we add to the existing literature and contribute to the understanding and implementation of GBL in higher education. The paper primarily provides useful insights to lecturers interested in introducing the GBL method into their teaching process to increase student motivation and engagement in the learning process.

## 2. Materials and Methods

The paper results from a two-year international project involving five European universities (i.e., the University of León, University of Ljubljana, Dokuz Eylul University, University of Gdansk, and Heilbronn University of Applied Sciences). As part of the project, we used an instructional design approach to develop a comprehensive process procedure for implementing a cost-effective GBL method in higher education. Specifically, we first analyzed the existing literature in the field of GBL and the experiences with using the GBL method at the aforementioned universities. Then, we analyzed the main characteristics of the students and the digital competencies of the teachers and students. In the next step, we defined the learning objectives we wanted to achieve with the GBL method. Based on the collected data, we developed the GBL method. The GBL model was then tested and evaluated at five universities. Based on the literature review and the experience we gained in developing, testing, and evaluating a specific GBL method, we developed a comprehensive step-by-step guide on implementing GBL into course didactics systematically and cost-efficiently. As part of the project, we also identified security, cultural, and quality assurance issues that need to be considered when implementing GBL in higher education [26].

## 3. Results

### 3.1. Processes and Steps for the Comprehensive Implementation of the Game-Based Learning Method in Higher Education

The number of publications on gamification and GBL has grown rapidly in the last ten years, from only one publication in 2012 to about 1200 publications by 2021, describing experiences in various fields on how to introduce game elements into the learning process to capture students' attention [27,28]. Lecturers who want to introduce the GBL method into their classrooms have to study the ever-growing GBL literature, which is simultaneously fragmented, operationalized, tested, and applied in numerous ways. The introduction of new methods requires much time and energy that lecturers often do not have. In addition, while the goal of GBL is usually easy to articulate (i.e., to achieve a specific learning objective), the activities required to implement GBL are usually more time-consuming than expected, especially if there is no guideline available.

To design a quality and meaningful educational game, Pan et al. recommend a sequential process to achieve the objectives of the GBL [29]. The sequential process gives structure and milestones to the activities and turns them into some sort of project. This sequence also helps ensure that key activities are carried out appropriately and that we remember all important issues for overall success.

This paper presents a comprehensive process of introducing GBL in higher education, which includes thirteen steps to help lecturers introduce GBL into their teaching processes. This process aims to make GBL as useful and accessible as possible for a wide range of lecturers. The sequential process is organized into three subsequent phases: preparation, game design, and practicing and evaluation (Figure 1). In the following, the individual phases and the steps will be described in detail.

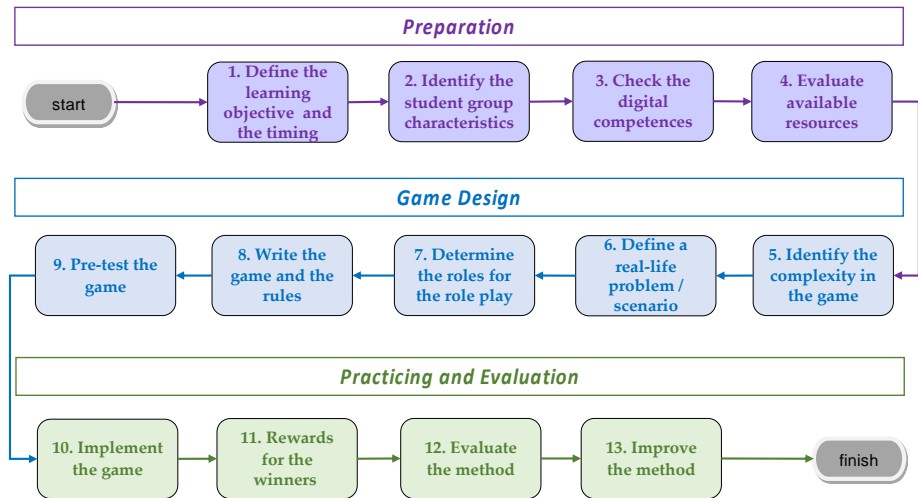

**Figure 1.** Graphical representation of the process and steps for the comprehensive implementation of the GBL in higher education. **Source:** own elaboration.

3.1.1. Preparation Phase

In the first step of the preparation phase, lecturers should identify and define the learning objectives and the competencies they want to develop through the GBL method based on the curriculum and the course of study. Setting clear learning objectives to be achieved through GBL activities benefits both students and lecturers. It is an important factor in determining the successful integration of games into the learning process [30,31]. When defining the learning objectives to be achieved through GBL methods, the lecturer should consider the students' interests and needs to ensure that the game is interesting and meaningful to students. It is also important to set realistic expectations in terms of objectives and competencies, depending on how much time lecturers plan to devote to game-based learning. The more time devoted to the developing GBL, the more content can be covered using the GBL method. At the same time, this also means that lecturers will have to spend more time preparing the content and related materials, which can significantly increase the complexity of the game design phase. This step significantly impacts the other steps, so lecturers should consider it carefully.

In the second step, lecturers should identify the particular characteristics of the student group in terms of size and other aspects important for teaching, such as the homogeneity of the student's knowledge base, multicultural or multilingual background, and the dynamics they experience. Although it is possible to conduct GBL activities with various groups, inexperienced lecturers are advised to start with small, preferably homogeneous groups of up to 25 students and to conduct GBL activities with more complex groups when they have gained more experience. It is also important to note that existing research suggests that group size can affect the impression gamification and GBL makes on students in terms of student interest, comparison, and discouragement but not in terms of student effort, perceived choice, perceived competence, stress, or motivation [32].

In the second step, lecturers must also determine the modality of the game. That is, whether GBL activities should be conducted only in person or face-to-face [33], exclusively online [34], or in a hybrid way [35]. All modalities have advantages and disadvantages. In general, the existing literature suggests that the hybrid modality is the most recommended, as it allows students to show engagement in the face-to-face sessions but under the neutrality and competition of the online modality. Synchronicity creates a positive learning synergy after conflicts within the formed teams, which is reinforced by the GBL process between groups based on a competitive strategy [36]. Suppose lecturers want to conduct GBL activities fully or partially online; in that case, they need to evaluate the digital competencies of themselves and students, as well as the possibilities of access to the Internet and computers or cell phones that allow the introduction of online or hybrid tests.

This leads us to the third step. Defining the learning objectives and the competencies, together with determining the modalities of the game, leads us to the third step. In this step, lecturers should consider the digital literacies of both students and lecturers, as the level of digital literacy of key stakeholders significantly affects the development of the GBL method. Students should have the digital competencies to use the game and effectively engage with the learning materials. Lecturers, too, should have the digital competencies necessary to effectively integrate the game into the curriculum and support students as needed. At this point, it may be necessary for both students and lecturers to undergo a realistic competency check and, if needed, training to improve their digital competencies. The common European framework (Digcompedu) could be used [37] should be used to assess lecturers' and students' digital competencies. This system establishes free tools offered by the European Community to classify citizens into four levels of development: basic, intermediate, advanced, and highly specialized digital competencies. If a low level of digital competencies is identified, it is recommended to train the required competencies, for example, by using free courses available at the University of León [38].

To complete the preparation phase, lecturers should assess the resources available for GBL activities in the fourth step. In this step, lecturers should consider the possibilities offered by the university and other free and accessible resources (e.g., applications, websites, platforms, servers, Moodle) that can be used to implement the GBL method.

What are the minimum requirements for doing digital GBL? A reasonably efficient Internet connection is usually sufficient for most scenarios. Modern terminal devices such as computers, tablets, or a smartphone, with an up-to-date processor and a high-resolution screen, are sufficient to enter the world of GBL.

The setup mentioned above for starters in GBL will likely not be sufficient. The situation changes when the goal is to set up a more sophisticated type of GBL, such as virtual reality games (VR) or simulations in science, engineering, and business. Such games can be very resource and budget intensive. They generate a large amount of data that must be processed, calculated, and sent through the connections. Thus, to perform some advanced GBL activities, additional monetary resources are required to purchase additional hardware, software licenses, and a budget for software developers and supporting IT specialists.

### 3.1.2. Game Design Phase

After finishing the preparation phase, the actual material design of the game begins, where the rules, characters, and flow of the game will be established. According to the previous phase, several elements have been defined: objectives, timing, modality, number of students involved, the level of digital knowledge of the lecturers and students, and the availability of a budget, which will be essential when deciding on the complexity of the game.

Therefore, the fifth step can be started, where the game's complexity will be outlined.

According to the authors' experiences, when moving into game-based learning, one should start with a minimum of complexity. Only a small part of the class's learning content should be turned into a gaming experience. Gaming elements such as quizzes, puzzle games, or even board games such as the so-called "Beer Game" for teaching students about supply chain management, as described by Martinez-Moyano et al. [39], can be used in

the first approach. If the game is to be played online, the equipment described above (the fourth step) should be sufficient. Lecturers can easily use such games without deeper knowledge of game development and programming. Another advantage of this minimum approach is that the existing lectures do not have to be changed significantly, as the games are just an add-on to the existing teaching content.

If the first cycle of lectures with gaming elements was successful, the complexity could be increased for the next semester. Many games are also available on the Internet, and many are free. As the quality and professionality of these games vary greatly, we recommend checking them thoroughly regarding the technical quality and suitability for your goal-setting, as described in step six. It is also essential to ensure that the games are integrated into the course of the lectures. Games can replace or deepen teaching content. The games are now an essential part of the semester or block course and not only a supplement to traditional classroom instruction.

The next stage of complexity would use games as the main part of the lectures. The game is now an essential part of the semester or block course and not only a supplement to traditional classroom instruction. The theoretical part would consist of teaching a class to lay the groundwork and enabling the students to understand the game's logic. Such kinds are much more sophisticated and might not come free of charge. If developing self-designed games is not an option, purchasing a license for a professional game might be the best idea. Additional support from IT specialists may be required to set up and host the game server, as the requirements for the IT infrastructure will be higher. However, if the choice is to use a commercial game solution, the use of the server is included, as all data entry and calculations are completed at the vendors' facilities. The vendors usually also provide training and support for the teacher on how to conduct their games.

In all cases, it must be ensured that the game's current score is saved in the event of an interruption, such as a loss of the Internet connection or a power outage. It must also be ensured that the game can be continued at the point of recovery without any data loss.

Once the technical and budgetary requirements are known, the next step, number six, will define the real-life problem/scenario and the story environment. Again, the available resources and funding play an important role because more activities require more resources [40].

There are several very attractive topics for students from game fantasy, such as antiquity, the Romans, the Egyptians, medieval times, space, the center of the earth, or underwater life. However, much more important is to analyze the knowledge required from the students. The approach will differ from a stem or social science perspective. However, in any case, it is essential to adapt the challenges to the student's knowledge. The main reason is that if students feel they cannot solve a challenge, they will abandon the game, and the potential advantage of game-based learning will be lost. The background knowledge in STEM area subjects (science, technics, engineering, and math) will include math, physics, or chemistry, which may be needed to solve the challenges. Without a previous level, it will be challenging to be able to engage students. For social science subjects, adapting the vocabulary and story to the mean level of the class will be the key to involving the students.

Another important aspect is to incorporate a wide variety of activities, to ignite students' curiosity in different ways, and ask them to use their different intelligence. It is also important to think about how presenting these challenges helps to motivate students, through content, audio, and aesthetics, which are spheres that stimulate sensations and, therefore, the engagement of the students [41].

The seventh step would be to determine substantive roles for students, keeping in mind that they should be well-defined, challenging, interesting, specific, and designed to stimulate student learning. The objectives and the story will define the number of roles and the number of students who will play the same role. It is important to design them so way that there is competition and conflict in playing each role (a fundamental element of GBL). The example may be competing with another participant in the same role for the

best position. In addition, this step can enrich the design in several ways. One of them may be adapting the game design to people with different disabilities, designing the activities in an international setting in teams of other nationalities or disciplines, or even involving lecturers from different but complementary disciplines. In this step, it is important to consider the various codes of conduct and interests of the population groups. Once the roles are established, use GBL elements (e.g., Gamification Canvas model, Hero's Journey model, or other models) when designing the roles.

In the eighth step, all the elements we have already decided upon are united in storytelling, which is an engaging narrative of events, with a final message that leaves a learning or concept. To better connect with the students, it is beneficial to complete a language editing of the story in this step to adapt the story's vocabulary to the vocabulary used by the students, allowing them to identify with the situation more easily. The sequence of activities presented in the game must have logic and a significant learning process, and the difficulty of the different tests must be increasing [42]. It is also necessary at this point to consider what support materials the students will need to be able to solve the challenges at each step. Likewise, it works very well that the previous tests give the players access to gain power, valuable tools, or knowledge to compete in better conditions in future challenges [25]. The wording of the rules, points, and prices can be crucial to continue the game, but because of the learning objective, students must have many chances to continue in the game so that they do not drop out. Rewards depend on the creativity of the lecturer [43].

The ninth and last step of the game design phase consists of pre-testing the game to validate the game. For this step, it is crucial that the rules are clearly explained, and the measurement of this is done independently by the comprehension ability of each student. This phase is done in two ways. On the one hand, a few students are selected to play a pilot version of the game to analyze the flow of the story, whether the game captures them or not, improvements in the rewards of the phases, the possibility of adding new phases, tests, opportunities, resources or even the modification of the roles and scenarios. In addition, on the other hand, a small recording of the game will be made, and together with the rules, it will be shown to a group of experts with much experience in GBL who will validate its interest in teaching.

### 3.1.3. Practicing and Evaluation Phase

After revising the game, it is time to evaluate the correct performance of the game as an activity of the academic course.

Again, some steps must be followed, starting with the tenth step of implementing the course in class. Under the supervision of the lecturer who has designed the game and explains the rules of the game (which must remain written at some point of consultation by the students), the students start the game. It is important that automatic feedback channels are enabled to generate a good rhythm in the game and that students know when they have done it right.

The eleventh step requires designing a system for assessing the player's progress, where all players receive small, consistent positive reinforcements while identifying the winners. This way, it is possible to use this game to evaluate the students. The rewards will be part of the final evaluation.

The twelfth step consists of an evaluation of the method by the users of the method (the students), who must evaluate their learning, their assessment, and their engagement, as well as make proposals for improvement in the game. Additionally, the lecturers in charge of directing and designing the activity must evaluate whether it leads to better learning of the competencies and contents by the students. With all this information, lecturers can define new possible improvements in the design, and consider new roles or ideas to be introduced for subsequent editions.

Finally, in the thirteenth step, all these ideas are incorporated into the game to improve the game, as well as small activities that serve as indicators of its quality. It is very important

to carry out these phases of consolidation of the lessons learned to achieve a functional, fun, and successful learning design for the students. After the experience, we can even consider regrouping the students in different ways, looking for more competition among equals, turning the individual game into a group game, or using all the variables we can think of.

### 3.2. Environmental and Organizational Aspects of the Proposed Process

3.2.1. Security, Protection, and Cultural Issues in Game-Based Learning

This part concerns how to change curricula and involve the necessary security and cultural aspects when introducing game-based learning to overcome and solve potential challenges within this process. It provides detailed information on how to design GBL programs in an engaging and secure way. It covers topics such as data protection, fair play, and participants' privacy, together with the cultural aspects of GBL.

Security-based issues are important aspects of implementing GBL into the teaching process. At the initial stage of the course, it is recommended to plan the change of curricula and syllabi, which brings the further involvement of necessary formal aspects to the aims and functions of the GBL process [44,45]. Farooq et al. and Grijalvo et al. noticed how challenging it is for the lecturer to create the environment and terms in which students will be independent students [41,46]. The benefits of GBL depend on the security and comfort of the final users, which create potential challenges for a lecturer and an institution. Among the essential security aspects, there is also data security planning when introducing game-based learning. In this context, all the elements of GBL courses shall be planned in compliance with General Data Protection Regulations (GDPR) [47].

There are a few key things to consider when using GBL in a secure manner, which includes protection and ethical considerations, as well as practical tips for designing and implementing GBL programs. These elements are policy for data security, choice of an e-learning platform, password protection and email address recognition, protection and security of a device, and protection of personal data.

Before starting the GBL course, it is recommended to evaluate the security recommendations for lecturers, both existing at the higher education institution and considering the specific course. It is recommended to check whether the university has adopted a policy for data security in online learning and follow the adopted procedures. University policies for data security in online learning may help lecturers to conduct online courses more safely and securely. Many HEIs have adopted policies for data security and protection.

Secondly, suppose the university supports or recommends a specific e-learning platform as a learning management system for delivering online courses. In that case, it is good to use it to support students with the game resources and to deliver the midterm or final results [48].

It is then important to store data, materials, and results safely at password-protected sites at university-approved platforms. The most often used tools for sharing files regarding courses provided for students are available and used mainly in learning management system platforms and university websites: Moodle, Zoom, Microsoft Teams, and Google Meet. Some academic lecturers share the files with students via Google Drive, MS Teams, OneDrive, or other methods [44]. The common security mistakes practiced by game users are related to password creation and security. The users should be encouraged to choose strong (computationally complex), memorable, and unique (not re-used) passwords, which are important for game development. They also should use different passwords for different game-based courses.

It is necessary to access the course through an approved email address. For security reasons, it may be important to avoid using private e-mail addresses based on other platforms and use the university email address, which applies both to a lecturer and the students. Some university lecturers can use their private email addresses to contact students regarding course-related matters, which is also mentioned in the literature [49]. Access or authentication to the game platform should be associated with a high level of security. Unauthorized users cannot gain access to the content, the game, or download

materials reserved for course participants. In most universities, students are automatically enrolled by the university in e-learning platforms, and they use a university username or university email and password to log in. These methods of enrolment and logging in are considered desirable for security reasons, assuming that relations in game-based learning are multi-dimensional. Ellis et al. described four perspectives of interactions: student-content, student-lecturer, student-student, and student-interface [48]. Nevertheless, careful consideration should be given to other methods of enrolment and logging in, for example, when students create their accounts by email-based self-registration or their accounts are created manually by lecturers responsible for courses [50].

Another security aspect is the protection and security of a device. Lecturers often use private portable computers or other mobile devices to provide online courses. Still, in such cases, it is important to use a strong password to unlock a device and not disclose the password to others. Using a strong password is crucial for keeping a device and data safe. Using any device to deliver online courses should be aligned with protecting it with up-to-date antivirus software. Installing and keeping up-to-date antivirus, firewall, and malware software are fundamental security requirements, which is a weak point of many systems. Some higher education institutions also supply lecturers with antivirus software for private computers. The protection and security of a device also concern updating the operating system [46]. Operating system developers usually release operating system updates regularly until they decide their product is unsupported. These updates often contain security patches and new security features, which are important to install.

The next issue of security is the use of a secured Wi-Fi network. Wireless network security is vital to protect data from unauthorized access. Following the institution's regulations towards placing documentation related to online courses is a very important aspect of security and privacy protection, as well as following the rules on how long and where to store documentation related to online courses. In some HEIs, documentation can be removed immediately after the end of the course. In others, documentation needs to be kept longer than one year. Similar security and privacy protection rules may apply to sharing documents or other materials (including meeting links) planned in the course to work with external parties.

The rules regarding taking screenshots and recording any of the material or communication used during GBL courses are also the subject of privacy, security, and general data protection. It is important to inform all the game-based course users about the rules regarding taking screenshots, recording any part of the material, using voice communication during online classes, using a camera or an avatar, the methods of communication with a lecturer, etc. These rules may vary considerably at HEIs. Some systems will automatically delegate the form of an avatar or create a virtual identity to a user within the game mechanics [45]. Protection of personal data is, however, necessary. GBL often involves collecting and storing personal data, such as names and scores. It's important to ensure that this data is protected and stored securely, in compliance with relevant internal and external regulations.

Other important aspects in gamification and Game-Based Learning are cultural issues. The results of gamification or game-based learning courses are indicated in most papers as very optimistic and come with the greater involvement of the students. It is crucial, however, to incorporate formative evaluation to evaluate a student's performance with a good personal identification of a player in the system. GBL should be designed to be fair and equitable for all participants concerning personal protection and cultural respect. To provide recognition, the student may use software for avatar creation, which helps to participate in a game with a proposed avatar feature. Lecturers can strengthen appropriate actions by awarding points to a student's avatar, which is neutral and can be publicly viewed by other students. On the other hand, it also helps identify and avoid manipulating scores or outcomes or unfairly rewarding certain participants.

The lecturer may encourage safe behaviors and monitor them subsequently. Using game mechanics such as points, rewards, and leaderboards, the above-mentioned security

elements may be supported and incentivized [51,52]. The rewards and immediate feedback, are crucial for learning and engaging. The other important factors are randomness, automation, discovery, emotional entailment, playfulness [43], learning efficiency, workload, satisfaction, motivation, and enjoyment [45]. The lecturer may also come through safety training, or create reminders for course participants, to make procedures more interactive and engaging, increasing the chances that students will pay attention to such elements.

Aries et al., or Parra-Gonzales et al. state that GBL can be a powerful tool for engaging and motivating students, both stem and social studies and at different levels of education [53,54], but it's important to consider cultural differences when using it. The issue is to sustain cultural sensitivity in GBL, which signifies the importance of considering cultural differences and using the modes of designing GBL programs that are effective and appropriate for participants from different cultural backgrounds.

The aspects crucial for using GBL in a way sensitive to cultural differences cover respect for cultural values and beliefs, using universal game mechanics in the form of rewards and symbols, and ensuring effective communication.

Different cultures' representatives may have different values and beliefs that influence their attitudes to the GBL process [55]. Grijalvo et al. and Manzano-León et al. emphasized that some cultures may place a greater emphasis on teamwork and collaboration. In contrast, others may regard individual engagement and achievement more highly [41,56]. Research findings showed areas of great concern: subjectivity in definitions and the linkage between GBL and motivation factors. User experience is described in most studies through autonomy, competence, and relatedness while playing [51,52]. Some studies show, however, that GBL may have a negative impact on learning math and collaboration or preferences to work in teams [57]. It is important to consider these differences when designing GBL programs to ensure that they are appropriate and effective for all participants. A good practice may be inviting foreign users to log in to the platform to check the design and cultural features of the game before its launch. Such experience was mentioned by Metwally et al. in the case of English lecturers who expressed feedback towards the language learning platform [50].

As a base, GBL typically involves using game mechanics such as points, rewards, and leaderboards that are very well described from the point of view of game structure [41,51,56]. These mechanics should be universal in meaning to be understood and appreciated by people from different cultures. By using these universal mechanics, course lecturers shall create GBL programs accessible to all students, regardless of their cultural background. It concerns all parts of the GBL system, which are mechanical elements (such as cards, gambling, trade, attack, competition, and cooperation); dynamics (behaviors such as socializing, reflection, status, and attention); or aesthetics (emotional interactions such as narrative, challenge, expression, and entertainment) [41]. There are many examples of game-based courses which allow the participants to become acquainted with cultural heritage, including art, music, history, and geography, in different countries, such as China, Japan, European countries, and other places. The advantages of such games are twofold: learning language through an immersive experience and enhancing awareness of cultural differences. Some studies show the importance of participation in games or game-based courses, while being anonymous to other players. It is more comfortable for people who do not like challenging others but also for those representing cultures associated with little acceptance of failures or making mistakes or with a high value of success in the group or society [51].

Clear and effective communication is also important when using GBL with students from different cultural backgrounds. It is crucial to ensure that all of them understand the GBL course's rules, goals, and rewards. The lecturer should plan information delivery in a way that is accessible and easy to understand. Adapting to GBL, the concept used to denote communication interpretation of the environment, called communicative staging, we may see the direct reference to conveying experiential meanings from service provider to customer in the companies and from lecturer to the student at HEIs [40]. The manner

and content of the communication may be reflected from students to lecturers and between students themselves. Its efficiency relies on the mutuality of cultural understanding.

Lastly, gamified courses may also focus on teaching students topics such as information security awareness, introduction to computer security, digital forensics, game development, diversity, and gender or cultural awareness. Concerning the contemporary challenges, there are suggestions of involvement in university education GBL for sustainable learning and building pro-environmental behavior [56].

GBL programs should be regularly evaluated to ensure they align with the course curriculum and HEI's values, which helps avoid unintended impacts or consequences.

### 3.2.2. Quality Assurance for Game-Based Learning

Quality assurance (QA) is defined as planned and systematic actions carried out within the quality system and demonstrated as necessary to ensure adequate confidence that a product or service will meet specified quality requirements [58]. The focus of QA is directly on quality-related processes and product outcomes [59]. From the standpoint of HEIs, quality assurance of an academic program gives students, lecturers, and other stakeholders' confidence that academic quality requirements will be met and the academic program will serve its intended purpose, i.e., providing high-quality education [60]. In terms of course quality, there has been a shift in recent years to a system that focuses on process and considers a combination of factors that contribute to the educational experience and learning, as stated by Tariq et al. [61]. These factors include students' needs, data, information used for decision-making, administrative input, and learning outcomes improvement [62]. The quality of the online learning environment is firmly based on the pedagogical needs of the course and its students, being reliable and robust, aligning with the technical infrastructure of the institution, and being regularly subjected to internal evaluations, updates, and improvements as needed. In a recent study, Hafeez and Sultan (2022) explored the notion of quality, quality assurance indicators, and models used to analyze and assess the quality of online learning [63]. According to the study, the most important indicators of quality assurance are student–lecturer interaction, documented plans for implementing technology in online learning, a student support center, and evaluation and assessment of the quality of online learning programs. Specific to the evaluation of learning games, Dondi and Moretti established an evaluation framework, which made a framework to support self-evaluation and defined the Sig-Glue quality criteria for learning games [64]. The general remarks of these assumptions are presented as follows:

- Any digital resources used in learning and teaching processes must meet quality criteria in methodology, context, content, and technology. The significance of each area is determined by the overall design of the learning experience and the role assigned to the resources themselves.
- Quality assurance in game-based learning is critical because some cultural, psychological, and social resistances can only be overcome by demonstrating that the games are 'serious', 'reliable,' and 'effective' in supporting the learning and teaching process.

Considering the findings of the studies mentioned above, there are more discussed common aspects of a quality experience in the traditional and online learning environment. The diversity of the existing knowledge base and practices can generate confusion in selecting the most appropriate criteria for QA. Instead, it is reasonable to consider two essential prerequisites: identifying the quality characteristics sought (inputs (e.g., entry standards and staff qualifications), processes (cycle time for an enrolment process or time to receive feedback from assignments), outputs (completion rates) or outcomes (knowledge and skills acquired, including life-long learning skills), and assessing attainment (quantitative measures or qualitative judgments or both). Then, based on the context in which GBL is implemented, a customized QA framework can be formed. As was noticed by Abdous, understanding the context frames for the QA process dictates the standards used during that process, particularly when translating the existing standards into operational checklists [65].

Following Vagarinho and Llamas-Nistal, for the GBL method, a process-oriented QA model [66] is offered. This model includes three sequential stages in line with the process model of game-based learning proposed in this study and is consistent with the outcomes of Chapter 2. The stages of the model comprise Preparation, Game Design, Practicing, and Evaluation. From the perspective of QA, the first stage refers to the upfront planning and analysis, which presents the basis for subsequent stages. This step includes environmental scanning to understand the context in which the Game-Based Learning method operates. The outcome of this scanning is the identification of the learning objectives, the key stakeholders (students, lecturers, administrators, IT personnel, etc.) and requirements, the competencies, the resources to be allocated, and the overall infrastructure.

Subsequently, during the stage of game design, the main activities focus on identifying game complexity, defining the scenario, roles, and rules, and validating the game design. This stage creates the actual output of the game, which will then be assessed for its performance in the next stage. The third stage corresponds to the post-production of the GBL method. Activities falling in this category can include stakeholder feedback about the GBL method, performance evaluation, and assessing the consistency of the GBL method concerning the learning objectives and the overall performance with the expectations of all stakeholders. After consolidating learning and stakeholder feedback, the lecturer can continually use this information to improve.

During all process steps, QA checklists, which can be drawn from evidence from research-based standards, could be used by lecturers or instructional designers to ensure the application of the standards and guidelines identified throughout the process. Some predesigned content collection templates can be used to ensure the content's appropriateness, comprehensiveness, and consistency. These templates should include the critical elements of the subsequent steps of the GBL process. Once created, such tools are useful for self-assessment and improvement as well as for fostering a culture of quality since they facilitate QA implementation, monitoring, and reporting. For this study, some predefined questions were adapted from Fotaris and Mastoras to the process preparation and game design stages presented in Table 1 [67]. Timbi-Sisalima et al. also published a guide for the self-assessment of the quality of accessible virtual education, one of the most comprehensive self-assessment models available in the literature [68]. Thus, internal feedback mechanisms for all components of the GBL method's QA system can be developed using comprehensive self-assessment models as a reference.

**Table 1.** Checklist Questions for Game-Based Learning Process Tasks.

| ID | Process Steps | Checklist Questions |
|---|---|---|
| | | PREPARATION |
| 1 | Define the learning objective and the timing of the game | What is the overall purpose of the game-based learning (GBL) method? What are the learning objectives this GBL is going to support? Where will the GBL be positioned in the course curriculum (e.g., as a stand-alone activity, at the introduction of a course, during a course in addition to a lecture, as an assessment, or as a serial story)? How many learning outcomes are sufficient without overloading participants? Why is achieving each learning objective significant? What will be the duration of the game? |
| 2 | Identify the student group characteristics | How many participants need to play at the same time? Will small groups play the game, or does it need to be scaled up? How many sessions will be necessary to involve all participants? |
| 3 | Check the digital competences | Do students have the necessary skills to play the game? |
| 4 | Evaluate available resources | What type of GBL will be developed (e.g., physical, digital, hybrid, etc.)? If the GBL is physical, where will it be located (e.g., outdoors, in a classroom, lab, library, office, etc.)? How much staff time do you have available to run the activity? Does the goal require the right amount of effort? |

**Table 1.** *Cont.*

| ID | Process Steps | Checklist Questions |
|---|---|---|
| | | Is there a sufficient budget to develop the GBL? |
| | | Are the necessary resources available (e.g., space, props, equipment)? |
| | | What is the deadline or time restraint in developing the GBL? |
| | GAME DESIGN | |
| 5 | Identify the complexity of the game | Will you develop alone, or will you co-create with the target audience? |
| 6 | Define a real-life problem/scenario | What knowledge is required to succeed in the game? Is it explicit, assumed, retrievable, or a mix? |
| | | Will the story be stand-alone such as a full movie or framed as an episode with a continuous narrative arc? |
| 7 | Determine the roles for the role play | Are there any tasks that may prevent participants with differing mobility levels or sensory impairments from playing the game? |
| | | Are there any language barriers that may prevent non-native speakers from playing the game? |
| 8 | Write the game and the rules | How will the game be monitored? |
| | | Will the GBL be used as a formative or summative assessment tool? |
| | | How will the designer know when the game is successful? |
| | | How can you quantify or qualify the learning objectives that have been met? |
| | | How much time will be available for self-reflection after the game? |
| 9 | Pre-testing and validating the game | Do experts validate the model? |
| | | Are there any other possibilities of qualifications? |
| | | Will this game be played with different players? |
| | | Are there any new possibilities for adding more resources? |
| | | Are there any better ideas or new roles to incorporate into the game's storytelling? |
| | PRACTISING and EVALUATION | |
| 10 | Implement the game | Do students understand well how to play their roles? |
| | | Do students know where they went wrong? |
| 11 | Rewards for the winners | Do students know about their progress? |
| | | Are students motivated by the reward? |
| | | Do students know who the winner is? |
| 12 | Evaluate the method | Do students give feedback or suggestions for the game? |
| 13 | Improve the method | Are there any opportunities to improve? |
| | | Is there any reason for revising? |
| | | Are there any lessons learned to improve the game? |

Source: Checklist questions were adapted from Fotaris and Mastoras [67].

## 4. Discussion and Conclusions

A key role of HEIs is to provide students with the knowledge and competencies needed in the job market. To achieve this goal, HEIs must be flexible and adapt quickly to changes in the wider environment. Digitalization trends and the COVID-19 pandemic have significantly changed the organizational and technological aspects of HEIs functioning and increased the need to change the pedagogical approaches of lecturers [5]. In this paper, we have focused on describing a detailed instructional design to promote the implementation of an innovative pedagogical method (i.e., GBL) suitable for teaching digital natives who have unique characteristics and needs [15] and that can be used to promote the achievement of learning objectives while equipping digital natives with the necessary knowledge and competencies for employment, decent jobs, and entrepreneurship, which is also a requirement of sustainable development. The research shows a high level of compliance with the Sustainable Development Goal (SDG4), which is dedicated to the quality of education and addresses information and communication technology for teaching and learning as defined by the United Nations [69]. SDG4 promotes an active learning process, which aims to develop competencies for helping citizens to enter into action: designing images, searching for answers, and investigating, planning, and acting [70]. This type of GBL encourages students to be self-directed learners by presenting them with problems and questions they should solve independently. Most of the time, multiple solutions are available, as they might face in their future life and employment.

Specifically, we developed a comprehensive process for lecturers to follow; discussed security, protection, and cultural issues; and presented a process-oriented quality assurance model for implementing GBL in higher education.

Thus, the paper makes several distinct contributions to the GBL literature and the application of the GBL method in higher education. First, we contribute to the ongoing debate in the GBL literature by suggesting a step-by-step cost-effective process for lecturers interested in adopting the GBL method. In doing so, we also contribute to the mitigation of the challenges that lecturers face in adopting the GBL method (e.g., lecturers' lack of time to study extensive literature, lack of knowledge on how to successfully implement the GBL method, lack of (financial) resources, etc.) and thereby increase the prevalence of the method in higher education. Namely, despite all its advantages [22], GBL in higher education is still not widespread. According to the available research, the reasons could be that the method is quite expensive and challenging to implement, thus requiring a great deal of time and effort from lecturers [20]. Moreover, in the last ten years, the method has been mainly used by lecturers constantly looking for innovative teaching methods and introducing them into their teaching process to increase students' motivation and engagement in the learning process. Therefore, in the literature, we can find more or less only anecdotal evidence of the use of the GBL method developed for a specific learning objective in a particular subject, so it is difficult to generalize these experiences and apply them in another context. Consequently, the comprehensive and cost-effective process of introducing the GBL method presented in this paper contributes to a more systematic application of the GBL method in various science and social studies courses.

An important contribution of the proposed process is also that it is designed to be very flexible and therefore suitable for broad application, regardless of the type of course, the size of the group of students, digital competencies of students and lecturers, and the experience with the use of the method of the lecturers who wish to use it.

In addition, the process presented in the paper is designed to be extremely practical and provides practical guidance on how to implement the GBL method based on the experience of other researchers who have already carried it out. By describing the thirteen steps that lecturers should follow when implementing the GBL method, including the best practice and times to make revisions to the process, we contributed to the quality of the process, and we saved the time that lecturers spend learning the GBL method and applying it in their teaching.

The paper thus benefits lecturers, who can more efficiently and effectively incorporate the GBL method into their learning process, thereby enriching it. In addition, it also benefits students, as the implementation of the GBL method enables them to be more actively involved in the learning process, enjoy learning more, and thus increase their motivation for their studies. In addition, in the paper, we also pointed out the security issues and indicators that should be measured to accredit the quality of GBL.

We also contributed to the GBL literature by discussing the potential risks and benefits of gamification and GBL when used in an online environment and providing recommendations for best practices. The paper highlights the importance of balancing the benefits of GBL with the need to protect personal data and student privacy. It also discusses the potential benefits and challenges of using GBL in HEIs and highlights the importance of considering cultural differences. The paper suggests how lectures should adapt GBL activities to meet the needs and preferences of intercultural groups. GBL rules designed with sensitivity to cultural differences are essential for effectively engaging and motivating students from diverse backgrounds.

Finally, we contributed to the GBL literature by presenting a process-oriented quality assurance model for the GBL method. The importance of QA for various forms of pedagogical work became very clear during the COVID-19 pandemic when universities were forced to switch from face-to-face to online teaching and when lecturers and students encountered multiple problems, including the quality of online education [63,71]. During this time, HEIs have realized that they need alternative pedagogical approaches to promote

student motivation and engagement in such a situation. One of the promising methods for delivering online teaching and learning has been GBL, which has attracted much attention in recent literature. Among other things, GBL needs to be evaluated holistically and more from the perspective of students and learning dimensions to ensure its quality. However, there are no QA guidelines for this topic in the literature yet, as well as for security in gamification combined with a cultural approach. This paper, therefore, contributes to the GBL literature by answering the question of how GBL processes in higher education can be designed for QA. We argue that it is crucial first to evaluate the principles and procedures of QA in higher education to meet the fundamental requirements of GBL quality.

Although the number of articles on the GBL method has increased significantly since 2011, indicating that lecturers and the academic community are very interested in the GBL method, its use in higher education still needs to be improved. By developing a comprehensive procedure for implementing a cost-effective GBL method, we help lecturers in using the GBL method in their teaching and improving it as a result. Therefore, this paper may increase the likelihood that lectures will use the GBL method, impacting the student learning experience. Adopting an innovative teaching method and a positive student experience will, in turn, positively impact the attractiveness of higher education programs. The paper offers implications for the following stakeholders: lecturers, students, and HEIs.

## 5. Suggestions for the Future Research

Providing roadmaps is very useful for lecturers beginning their gamified classes and courses. The study identifies several opportunities for future research. The paper emphasizes the need for further development of competencies in gamification and game-based learning. Namely, the success of the implementation of the GBL method depends to a large extent on the competencies of lecturers and students. Future research could therefore investigate which competencies are required to successfully implement the GBL method. In addition, obtaining empirical evidence of how this method in higher education affects the development of transversal competencies would be essential for HEIs and study programs. To meet the demands of the job market, HEIs need to work systematically to improve their graduates' skills in teamwork, leadership, flexibility, innovation, and resource organization, which can be practiced in role-playing games [61,72].

Games can promote the development of these skills, so GBL can be seen as an appropriate approach that can be combined with traditional content delivery to achieve higher levels of needed skills. The GBL courses, if well-structured, equip students with interpersonal and resource-seeking skills. Piaget and Vygotsky [73] describe the development of competencies in cooperation with others.

Previous studies have mainly addressed the principles, mechanisms, and motivation of gamification or GBL. However, there is a growing need for a systemic approach to incremental GBL activities with careful consideration of quality and cultural issues. More and more degree programs have become internationalized, requiring a culturally sensitive perspective when introducing new teaching methods and technical advancement. Therefore, quality issues and cultural aspects should be included in a systematic approach to the educational process. Future research could evaluate the potential of using gamification and GBL in cross-cultural groups. In addition, future research could also examine the amount of time and effort lecturers must devote to conducting GBL activities within domestic and international courses and how HEIs should or could compensate for this.

Building a sustainable environment for universities starts with the values on which the courses are built. After the experience of being globally limited by COVID-19, HEIs are at a very valuable inflection point that has provided flexibility for both higher education institutions and the main education stakeholders: lecturers and students. Thus, a moment of change facilitates the incursion of these new GBL experiences into formal teaching. Enhancing the cultural and quality perspective may help to define the process, which will

be adjusted to the individual characteristics of the courses for different fields, stem sciences, and social sciences.

Future research should therefore address how this process can be designed to ensure the development of successful and functional educational experiences. In-depth reflection is also needed to develop an innovative and realistic proposal for assessing students' acquisition of competencies and content in GBL experiences.

**Author Contributions:** All authors contributed significantly to the final version of the manuscript. M.F.R. and D.A. coordinated the project and conceived the idea presented therein. M.F.R., D.A. and A.K.İ. carried out the analysis of the state of the art. M.F.R. proposed a process structure and A.K.İ. performed a deep analysis of it, designed the figures, and helped interpret the results. MM. analyzed security and cultural issues. M.F.R., D.A., A.K.İ., M.M. and H.S. drafted the manuscript, wrote—reviewed, and edited. H.S., M.M., M.F.R. and D.A. revised the manuscript. All authors have read and agreed to the published version of the manuscript.

**Funding:** This research was funded partially by the project "DigiMates, Development of Innovative, Gamified and Interactive Method for Advanced e-Teaching and E-learning of Skills" funded by EUROPEAN COMMISSION within Erasmus+ Strategic Partnership KA226 Partnership for Digital Education Readiness, grant number 2020-1-SI01-KA226-HE-093593.

**Institutional Review Board Statement:** Not applicable.

**Informed Consent Statement:** Not applicable.

**Data Availability Statement:** No new data were created or analyzed in this study. Data sharing does not apply to this article.

**Acknowledgments:** The authors would like to thank the "DigiMates" team, in particular, Susanne Wilpers and Susanne Stetter from the University of Applied Sciences Heilbronn, and also Adriana Suárez Corona from the University of Leon.

**Conflicts of Interest:** The authors declare no conflict of interest.

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
