# Peer review of "Development of a Comprehensive Process for Introducing Game-Based Learning in Higher Education for Lecturers"

_sustainability, doi:10.3390/su15043706_

Round 1

Reviewer 1 Report

Overall, this is a good research. The research introduced game based learning and introduces GBL in higher education and will help higher education to introduce such concepts. But on the other hand, the paper requires minor adjustments on the following comments:

1. What is the contribution of the study with respect to the Benefits and challenges for teachers and students, need separate discussion being the important points for the study.

2. The impact of study is missing as a paragraph.

3. Following new references should be added in the paper where similar methodologies and dicu are applied and by studying and adding these will improve the quality of this paper:

1.      Alfaifi, A. A., & Khan, S. G. (2022). Utilizing Data from Twitter to Explore the UX of “Madrasati” as a Saudi e-Learning Platform Compelled by the Pandemic. Arab Gulf Journal of Scientific Research, 39(3), 200-208. doi:10.51758/AGJSR-03-2021-0025

2.      P. Nikolaidis, M. Ismail, L. Shuib, S. Khan, and G. Dhiman, "Predicting Student Attrition in Higher Education through the Determinants of Learning Progress: A Structural Equation Modelling Approach," Sustainability, vol. 14, no. 20, p. 13584, 2022.

Author Response

Dear reviewer

Find Attached the answers to your comments.

Thank you very much.

Reviewer 2 Report

1. "Existing research suggests that game-based learning methods are expensive and difficult to introduce." I suggest a more detailed indication of which type of game. Because there are many free digital games that are widely used by teachers.

2. In the Introduction, I propose to introduce the theory of development for GBL comprehensive process.

3. A good Article should be based on data. Although this article proposes a new GBL comprehensive process method. But this article did not conduct an empirical analysis. Therefore, we cannot confirm its validity.

4. Please add the Materials and Methods, Results and Discussion

5. Please strengthen the relevance of this method to sustainable development in the conclusion.

In this article development concepts of GBL comprehensive process and instructions for use, are very clearly presented and can be seen to have high application value. However, I am concerned about the scientific nature of the development process. Therefore, I recommend rejecting this article.

Author Response

Dear Reviewer

Find attached the word with your answers.

Thank you very much for your time and your effort.

Reviewer 3 Report

Line 60: GBL instead of "GLB"

Lines 62-70: A pretty lucid explanation of the distinctions between game-based learning and gamification

Line 80: Perhaps: "On the other hand, a review of the literature on game-based learning also REVEALED (instead of "found") that some studies have not found clear evidence..."

Lines 96-98: This is a speculation or inference drawn from the literature?

Lines 148-149: Perhaps, this assumption is not necessarily correct: The more the time we spend in a process, the more the time we spend in the content.

Lines 163-173: It is ok. However, this paragraph could be another step or belong to the next step (third step)

Lines 192-193. I don't think that is necessary to mention a suggestion of speeds. In the next years this suggestion will be out of date.

Lines 194-195: The expression "if necessary" is not needed. 

Lines 221-222: As you also mention, it is not necessary for a teacher to be a programmer. There are several web tools (not necessary games) that can be used for GBL, that can be handled from everybody.

Lines 237-249: perhaps, it would be useful to mention the difference between High-end game-based learning and gamification.

Lines 212-249: In my opinion, the criteria (and their load) of the chosen categorization are not clear enough

Lines 255-274: Very useful step!

Lines 300-307: This step could also include the comprehension of the rules.

2.2. section: Perhaps, a "language editing" step would be useful.

3.1.1. section: A reference to GDPR might be useful

3.1.2 section: This is a very important reference

Lines 560-561: I suppose, you mean the context in which the Game-based Learning method operates

Line 594: COVID-19 pandemic is not a trend. Perhaps, you mean that it is a factor.

Lines 597-598: Actually, you do not introduce a new pedagogical method. GPL is a pedagogical method. The process you describe is some kind of an instructional design. 

Author Response

Dear Reviewer

Find attached the word with your answers. Thank you very much.

Round 2

Reviewer 2 Report

Dear Authors,

You have revised the manuscript and has improved significantly, so I will suggest editor that it can be accepted this revision.

Best regards